# Evaluation of Surface Change and Roughness in Implants Lost Due to Peri-Implantitis Using Erbium Laser and Various Methods: An In Vitro Study

**DOI:** 10.3390/nano11102602

**Published:** 2021-10-02

**Authors:** Aslihan Secgin-Atar, Gokce Aykol-Sahin, Necla Asli Kocak-Oztug, Funda Yalcin, Aslan Gokbuget, Ulku Baser

**Affiliations:** 1Periodontology Department, Faculty of Dentistry, Istanbul University, 34452 Istanbul, Turkey; aslihansecginatar@gmail.com (A.S.-A.); asli.kocak@istanbul.edu.tr (N.A.K.-O.); fyalcin@istanbul.edu.tr (F.Y.); 2Department of Periodontology, Faculty of Dentistry, Istanbul Okan University, 34959 Istanbul, Turkey; gokceaykol@gmail.com; 3PGG Private Practice, 34365 Istanbul, Turkey; aslangokbuget@gmail.com

**Keywords:** dental implants, titanium, laser therapy, peri-implantitis, debridement

## Abstract

The aim of our study was to obtain similar surface properties and elemental composition to virgin implants after debridement of contaminated titanium implant surfaces covered with debris. Erbium-doped:yttrium, aluminum, and garnet (Er:YAG) laser, erbium, chromium-doped:yttrium, scandium, gallium, and garnet (Er,Cr:YSGG) laser, curette, and ultrasonic device were applied to contaminated implant surfaces. Scanning electron microscopy (SEM) images were taken, the elemental profile of the surfaces was evaluated with energy dispersive X-ray spectroscopy (EDX), and the surface roughness was analyzed with profilometry. Twenty-eight failed implants and two virgin implants as control were included in the study. The groups were designed accordingly; titanium curette group, ultrasonic scaler with polyetheretherketone (PEEK) tip, Er: YAG very short pulse laser group (100 μs, 120 mJ/pulse 10 Hz), Er: YAG short-pulse laser group (300 μs, 120 mJ/pulse, 10 Hz), Er: YAG long-pulse laser group (600 μs, 120 mJ/pulse, 10 Hz), Er, Cr: YSGG1 laser group (1 W 10 Hz), Er, Cr: YSGG2 laser group (1.5 W, 30 Hz). In each group, four failed implants were debrided for 120 s. When SEM images and EDX findings and profilometry results were evaluated together, Er: YAG long pulse and ultrasonic groups were found to be the most effective for debridement. Furthermore, the two interventions have shown the closest topography of the sandblasted, large grit, acid-etched implant surface (SLA) as seen on virgin implants.

## 1. Introduction

Peri-implantitis is a pathological condition characterized by inflammation of the peri-implant mucosa and progressive destruction of the supporting bone as a result of biofilm formation on the implant surface [1,2,3]. Peri-implantitis is diagnosed by increased peri-implant probing depth, bleeding on probing in the peri-implant pocket, and detecting bone loss around implant with radiography [4]. Etiology of peri-implant infection comprises various factors such as the macro design of the implant, the level of surface roughness, and the condition of the hard/ soft tissue surrounding the implant [5]. However, bacterial plaque formation on the implant surface is the most important factor in the etiology of peri-implantitis [6].

There are multiple risk factors affecting the formation of peri-implanter disease, such as a history of periodontitis, poor oral hygiene habits, and smoking [7]. One or more of these risk factors may cause the degradation of the biocompatibility between the host and the implant surface [8]. The development of an adherent layer of plaque (biofilm) on the implant can result in the formation of calcified deposits similar to calculus and appears to be critical for the development of peri-implant diseases [9,10,11]. Therefore, the primary goal of treatment is to eliminate all microbial and calcified deposits from the surface of the implant and to supply appropriate conditions for re-osseointegration [8,10].

Despite the fact that periodontitis and peri-implantitis have similar features, they differ in some directions. These variations cause the treatment of the two diseases to own different ideas from every other [12,13]. Treatment methods such as tooth and root surface cleaning, which are effectively used in the treatment of teeth with periodontitis, cannot be used in the same way on rough and grooved implant surfaces [14]. In periodontal inflammation, it is possible to treat the infected cementum surface by cleaning it with various mechanical or chemical methods, whereas, in peri-implantitis treatment, successful surface debridement may not be fully achieved due to the complex implant surface topography. This creates an extraordinary shelter for bacterial attachment and colonization.

Increasing the amount of bone-implant contact and re-osseointegration of implants are vital for successful peri-implantitis treatment [15]. Numerous in vitro studies have analyzed the effects of debridement instruments used in peri-implantitis treatment which showed serious impairment on the titanium topography. Firstly, the traces that may occur on the implant surface impair cell adhesion to the titanium surface and damage proper wound healing. Furthermore, surface defects lead to changes in the implant surface characteristics, which may alter the bone integration of the implant [16,17,18].

Many in vitro studies evaluated the effects of decontamination and debridement instruments on implants by electron spectroscopy. According to the results of these studies, the most prominent pollutant was carbon. Carbon indicates the presence of any non-biocompatible substance (calcified and organic) that could change the re-osseointegration of implants. In particular, studies on failed implant surfaces have shown varying degrees of carbon [19,20]. Titanium-bone connection is altered by the deposition of organic molecules on implant surfaces, and it was concluded that the removal of hydrocarbon gradient is a crucial step in achieving the bioactivation and osseointegration of titanium [21,22]. It was also suggested that the implant surface must be treated to intensify the surface wettability and energy [23].

The high prevalence of peri-implantitis, ranging from 16 to 47.1%, has also prompted scientists to explore a range of therapeutic applications for implant surface decontamination [24,25]. These include mechanical debridement methods such as curettes, rubber cups, ultrasonic devices and powder spray systems, chemical disinfection methods such as chlorhexidine, tetracycline, metronidazole, citric acid application, and various surgical and non-surgical treatment methods including antibiotics, photodynamic therapy, lasers, and combinations of all these treatments [26,27]. However, the optimal procedure or procedures are still not fully specified [28,29].

Different laser systems have been tested for potential use in implant surfaces including erbium-doped:yttrium, aluminum, and garnet (Er:YAG), and erbium, chromium-doped:yttrium, scandium, gallium, and garnet (Er,Cr:YSGG) lasers [30]. Due to the high absorption by water, the 2.940 nm wavelength of Er:YAG lasers seemed to be capable of effectively removing bacterial deposits from either smooth or rough titanium implants without damaging their surfaces [31,32]. Although both Erbium lasers have high absorption by water, the Er,Cr:YSGG laser operating at a wavelength of 2.780 nm has a lower absorption coefficient in water than the Er:YAG laser [33].

Although the use of lasers is currently on the agenda, there is no standard recommendation regarding laser type, irradiation, or settings protocol for the treatment of peri-implantitis [34]. Before implementing a patient-level irradiation protocol, in vitro studies are required to establish ideal settings.

Our aim is to obtain similar surface properties and elemental composition to virgin implants using erbium lasers (Er:YAG and Er,Cr:YSGG) and mechanical methods (curette, ultrasonic device) in the debridement of implants lost due to peri-implantitis by evaluating the surface change of the implants treated with these methods by scanning electron microscopy (SEM), energy dispersive X-ray spectroscopy (EDX), and profilometry.

## 2. Materials and Methods

### 2.1. Sample Collection

Twenty-eight implants (DE2™, SLA G4) explanted because of advanced peri-implantitis, with debris on their micro-structured surface, and two virgin titanium dental implants (DE2™, SLA G4) representing the positive control group were included in this present study. Implants were autoclaved and immersed in acrylic blocs from a 3 mm apical part to facilitate the precise application of the instruments and standardization. SEM images, EDX, and profilometry measurements were done before and after debridement. Twenty-eight failed implants were allocated into seven test groups, and two virgin implants represented the control group. Test groups were divided based on treatment methods of implant surfaces and each test group had 4 failed implants. All assessments were done in two-time points, before and after the debridement. All implant surface debridement procedures were performed by the same investigator. This study was approved by the Ethics Commission of Istanbul University Faculty of Dentistry (Approved number 2020/22).

### 2.2. Interventions

Treatment procedures of contaminated titanium implant surfaces for each group are shown in Table 1.

In the titanium curette (Ti-Cur) group, the instrumentation was performed for 120 s by scaling in one direction at an angle of 30 degrees on the implants.

In the ultrasonic scaler with a PEEK tip (US-PEEK) group, debridement was performed by keeping the device tip in contact with the implant surface at a 30 degree angle for 120 s. The ultrasonic scaler was used at the recommended speed for routine periodontal treatment under maximum water cooling.

In laser groups, the laser energy delivery was directed by a computer interface that dictated the selected laser tip, modes, energy, and associated water and air for each laser group. Each used laser parameter was described in Table 1.

Er:YAG laser (ErL) (2.940 nm) was used with a 90 degree-irradiation angle to the titanium surfaces according to the recommendation of the manufacturer. An R02-C handpiece was used with a 0.9 mm diameter. The spot area was calculated as 0.63 mm^2^. To simulate clinical use, sweeping irradiation was performed in non-contact mode at approximately 1 mm [35,36,37].

Er: YAG very short pulse laser (ErL-VSP) parameters were: Pulse energy: 120 mJ; Pulse duration: 100 μs; Frequency: 10 Hz; resulting in an energy density per pulse of 19.04 J/cm^2^; Air/Water output: 4/6.

Er: YAG short-pulse laser (ErL-SP) parameters were: Pulse energy: 120 mJ; Pulse duration: 300 μs; Frequency: 10 Hz; resulting in an energy density per pulse of 19.04 J/cm^2^; Air/Water output: 4/6.

Er: YAG long-pulse laser (ErL-LP) parameters were: Pulse energy: 120 mJ; Pulse duration: 600 μs; Frequency: 10 Hz; resulting in an energy density per pulse of 19.04 J/cm^2^; Air/Water output: 4/6.

Er, Cr:YSGG laser (ErCrL) (2.780 nm) tip was used with a 15 degree-irradiation angle to the titanium surfaces according to the recommendation of the manufacturer. To simulate clinical use, sweeping irradiation was performed in non-contact mode at approximately 1 mm. An RFPT5 radial firing fiber tip with beam divergence > 40 degrees and 0.5 mm in diameter was used in the study. The spot area was calculated as 2.5 mm^2^ (0.025 cm^2^) at 1 mm from the implant surfaces [38].

Er, Cr: YSGG1 laser (ErCrL-1) parameters were: Power: 1 W; Pulse energy: 100 mJ; Pulse duration: 60 μs; Frequency: 10 Hz; resulting in an energy density per pulse of 4 J/cm^2^; Air/Water percentage output (%): 40/50.

Er, Cr: YSGG2 laser (ErCrL-2) parameters were: Power: 1.5 W; Pulse energy: 50 mJ; Pulse duration: 60 μs; Frequency: 30 Hz; resulting in an energy density per pulse of 1.8 J/cm^2^; Air/Water percentage output (%): 40/50.

### 2.3. Analyses

#### 2.3.1. SEM

Topographic surface alterations were assessed with SEM. The SEM observation was conducted with a FEI^™^ VERSA 3DLOVAC microscope (FEI, Hillsboro, OR, ABD). Analyses were performed at baseline for the control group, and before and after the interventions for the contaminated implants. Each sample was marked from the neck area to be observed from the same surface after the treatment. Each implant was scanned and photographed at five set magnifications (76×–150×–500×–1000×–2000×).

Implant Debridement Visual Index

This current index was created for the visual evaluation of SEM images. The index aims to compare the surface features of treated-contaminated implants with virgin implants by grading:
1:Image without any contamination and resembling a positive control,2:Spot contamination of observed image,3:Image of contamination beyond spot contamination.


SEM images of the implants after debridement were assessed at (150×) magnification. The debris on the implant surface were evaluated by three blind observers. The images were randomly shown to each observer twice without specifying the sample numbers and groups.

While analyzing by statistics, 1 and 2 grades were considered as clean, and 3 as contaminated.

#### 2.3.2. EDX

EDX analysis is utilized for quantitative evaluation and the local determination of the chemical composition. In the present study, EDX was used to measure the presence of carbon (C), titanium (Ti), oxygen (O), and nitrogen (N) elements on the implant surfaces. The spectroscopy of the emitted X-ray photons was performed by a Bruker detector with an energy resolution of about 123 eV at a working distance. The measurement was made from the 1 mm^2^ area on the same thread determined in each implant surface.

#### 2.3.3. Profilometry

The roughness evaluation and analysis were performed using a profilometry (Veeco Instruments Inc., Plainview, NY, USA, ABD), (Radius: 5 µm, Stylus force: 3 mg/29.4 µN, Resolution: 0.167 µm/sample, Length: 1000 µm, Duration: 20 s). Measurements were made at the same area where the identified threads in the middle third of each implant were marked. In the selected thread distance, the diamond tip of the profilometry performed measurement 20 times in the horizontal direction along the 1000 µm length and resulted in the average of the roughness.

#### 2.3.4. Statistical Analysis

A power analysis demonstrated a sample size of 4 implants for each intervention group would ensure 80% power to detect the difference between the treatment methods in the morphologic features of implant surfaces with a significance level of 0.05. The data analysis was performed using SPSS v.23 software (IBM Corp., New York, NY, USA). The descriptive statistics were presented by the mean values with minimum and maximum and their standard deviations (SD). When the data were parametric, paired sampled *t*-test was used in comparing the differences of the groups. For the evaluation of the before and after treatment differences between the groups, the statistical analysis used was one-way analysis of variance (ANOVA) and Tukey HSD multiple comparison tests followed by Bonferroni post hoc testing. The data are represented as mean ± standard deviation. A value of *p* < 0.05 was considered to be statistically significant.

## 3. Results

### 3.1. SEM Analysis

Implant Debridement Visual Index (IDVI) scores of the three trained observers are given at Table 2. According to the index results, the most effective groups were ErL groups. This was followed by the US-PEEK. The debris not removed (DnR) score was higher for ErCrL groups and Ti-Cur. All three pulse settings of the ErL were found to be effective in removing hard deposits from the implant surface.

All groups were contaminated at baseline compared to the control group. The contamination was confirmed by the presence of a layer of the debris, the titanium surface features, and by the EDX lower C and Ti values.

#### 3.1.1. Ti-Cur Group

Large and flat scratched areas were observed on the titanium surface after the treatment, in the Ti-Cur group. Although the honeycomb appearance [39] of the SLA implant surface was achieved in some surfaces, it was observed that the curette did not provide an effective debridement and left a residue (Figure 1a–d).

#### 3.1.2. US-PEEK Group

In the ultrasonic group, clean surfaces were observed similar to the positive control group. A small amount of debris and some materials considered to be remnants of the PEEK material were observed (Figure 2a–d).

#### 3.1.3. ErL-VSP Group

Although the debridement was achieved in the ErL-VSP group, delamination and deformation were also observed on the surfaces. In particular, porosity due to melting, loss of honeycomb appearance, and a relatively smooth surface with microcracks were observed (Figure 3a–d).

#### 3.1.4. ErL-SP Group

Although the cleaning process was completed, delamination, deformation, and melting were observed as seen in the ErL-VSP group. However, there were less undesired effects, and no microcracks were observed (Figure 4a–d).

#### 3.1.5. ErL-LP Group

In the ErL-LP group, debridement was achieved by the typical surface appearance of virgin implants at large magnifications (Figure 5a–d). The surface topography is comparable to virgin implant surface properties. No damage was seen on the implant surfaces.

#### 3.1.6. ErCrL-1 Group

In the ErCrL-1 group, it was observed that debridement was not totally achieved after laser application (Figure 6c). However, the microscopic appearance of some debrided threads was similar to the original nanomaterial surface, and no damage was observed on the surface (Figure 6d).

#### 3.1.7. ErCrL-2 Group

In the ErCrL-2 group, debris was remaining as a layer as in the ErCrL-1 (Figure 7c). The areas where debris remains are observed more clearly at 1000× magnification in the debrided threads (Figure 7d).

#### 3.1.8. Control Group

A nano-porous structure on the surface of the control titanium implants was observed. This nano-topography shows the typical micro-roughness of SLA implants (Figure 8a–d).

### 3.2. EDX Analysis

Higher C and lower Ti values were measured in the intervention groups at baseline [(Figure 9a,b) (*p* < 0.05, *p* < 0.05 respectively)]. All of the intervention groups were contaminated compared to the control group. The contamination was confirmed by the presence of a layer of the debris, and the titanium surface features of SEM analysis. C decreased (Figure 9a) and Ti increased (Figure 9b) as a result of debridement. While the highest percentage of C was found before the debridement of contaminated implants, the lowest C percentage was detected in virgin implants and in ErL groups after the debridement (Figure 9a). C contamination was significantly reduced after debridement procedures in all groups except ErCrL groups (*p* < 0.05).

O increased in the groups when achieving efficient debridement without surface damage (Figure 9c). According to our results, it is considered that the N also represents the contamination and decreased by debridement (Figure 9d).

### 3.3. Profilometry Analysis

Profilometry analysis results are shown in Figure 10a–p. A flattened three-dimensional topography was seen in the ErL-VSP group after intervention (Figure 10f). ErL-LP (Figure 10j) showed a surface topography similar to virgin implants (Figure 10p).

## 4. Discussion

Our aim was to ensure that the debris-covered dirty implants salvage the surface properties and elemental composition of virgin implants without changing the surface morphology after using different debridement methods. When SEM and EDX findings were evaluated together, the groups that were more efficient in re-achieving the typical nano-surface topography by removing the debris without damaging the surface of the SLA surface were ErL-LP and US-PEEK. Although ErL-VSP debrided contaminated implant surfaces more efficiently, undesirable outcomes were seen. The ErL-SP group caused some surface changes as well, and ErCrL groups, Ti-Cur could not remove the debris.

There are several in vitro studies evaluating implant surface properties after different interventions for debridement. However, these studies have some shortcomings. Firstly, most of the studies have been done with short-term biofilm formation on discs [40,41]. Studies that removed hard tissue residue from implant surfaces are limited. Removing a layer of biofilm or hard debris involves completely different interventions. There are few studies that removed the debris layer from the surfaces of implants that were extracted due to peri-implantitis [37,42,43,44]. As a result, definitive protocols for laser parameters do not exist. In this study, very short pulse, short pulse, and long pulse modes of ErL groups were used for debridement. The effects of different pulses on laser energy levels transmitted to a shorter pulse can have a stronger effect on the targeted area. Therefore, in vitro studies that evaluate the settings of lasers in terms of variant pulse modes along with mJ/pulse and time are vital for clinical application.

IDVI evaluate more objectively by comparing the virgin implant surfaces with dirty implants after debridement. The nanoscale honeycomb appearance of the SLA implant surface was assessed on SEM images (greater than 150× magnifications) of the cleaned implant surfaces. Scratching, melting, and carbonization on the implant surface due to debridement methods were ignored during IDVI scoring. Observers scored only debridement effectiveness (cleanliness) for the implant surface. However, the above-mentioned undesirable effects that occurred during the intervention were noted (Figure 1d and Figure 3d).

In our study, C, Ti, O, and, N elements were evaluated by EDX analysis. While C and N decreased after debridement, an increase was observed in Ti and O. EDX analysis around dirty implants showed that the lower percentages of C and higher Ti when the surface was cleaned [37,44]. In the studies of Scarano et al., increased surface oxide levels, decreased in porosity, and nano-roughness represented a positive change that could protect titanium against bacterial adhesion [43,45]. The study of Takagi et al. reported that C and Ca percentages on dirty surfaces decreased, while Ti percentages increased significantly after debridement with ErL and ErCrL on artificially created calcified areas. Substantial reductions in the percentage of O have also been reported. On the contrary, where there were natural calcifications on the lost implant surface, there was no substantial decrease in the O ratio in all groups [44]. In our study, the lowest O levels were seen in the ErL-VSP group, in which debridement was done thoroughly but some loss of nanostructure occurred. The absence of surface damage on SEM in the ErL-SP and ErL-LP groups made for interesting results of the study. The O ratio increased and reached levels similar to virgin implants in the two interventions. The element of O detected on implant surfaces can be attributed to the titanium dioxide (TiO_2_) layer, which prevents corrosion of Ti and increases biocompatibility. It was also reported that the thickness of the oxide layer on the implant surface could increase three to four times after implantation compared to pre-implantation [46]. Taken together, these findings suggest that the TiO_2_ layer remaining on the implant after ErL treatment may be important for healing in the later stages. In the SEM images of the ErL-VSP group, lower O percentages were observed in the areas where surface damage was seen. These results suggest that the measurement of elemental composition in addition to SEM images provides a quantitative assessment of titanium implant surface properties.

Hakki et al. reported that titanium curette was more effective than plastic, carbon, and titanium curettes [42]. However, when they compared the titanium curette with lasers, they reported that the curette left residue. According to the results of our study, similar to the results of Hakki and Takagi, scratches were detected in SEM images of the debridement areas (Figure 1d) [42,44]. Furthermore, the effectiveness of debridement was less than the laser’s application. The scratches that occurred by using curettes or ultrasonic devices on the implant alter cell adhesion on the titanium surface and thus effect proper wound healing. Harrel et al. compared titanium, stainless steel curettes, and PEEK ultrasonic tips in terms of metal particle release during debriding titanium implant surfaces. They reported that the PEEK tip had the least amount of titanium particles removed from the surface [47]. In this present study, more scratches and debris layers were observed on the surface of the Ti-Cur compared to the US-PEEK in SEM images (Figure 1c,d and Figure 2c,d). US-PEEK has been the most effective intervention group after the ErL in removing the hard debris layer according to the SEM and EDX analyses.

In the literature, few studies have performed debridement on failed implant surfaces that have been removed due to peri-implantitis [37,42,43,44]. In addition, the difficulty in comparing these research studies could be due to methodological differences as well as poorly reported laser parameters. When comparing laser devices made by different manufacturers, the optimum energy output differs between lasers. Therefore, it is very important for the clinician to fully understand the differences in the characteristics of different laser devices and to apply erbium lasers effectively and safely for the treatment of peri-implantitis.

When compared to debridement results in laser groups, ErL groups were found superior to ErCrL groups. Although the energy density of the pulse (19.04 J/cm^2^) in three ErL groups was the same, the applied pulse durations were different. All the ErL groups achieved debridement in the surfaces but two groups had some surface damage. ErL-LP debrided most effectively by achieving the typical surface appearance of virgin implants. It was considered that, because the ErL-LP group had the longest pulse duration (600 μs), the minor surface damages were in this group. On the other hand, although no damage was observed on the debrided threads, all the ErCrL groups failed to complete effective debridement. When the described parameters by the manufacturer were applied, it was calculated that the energy density of ErCrL-1and ErCrL-2 were 4 J/cm^2^ and 1.8 J/cm^2^, respectively. When comparing to ErL with ErCrL groups, the energy density parameters were lower in ErCrL groups. The inadequate energy density within the same duration of the application resulted in differing outcomes in terms of effectiveness among the interventions. Application duration is also critical for debridement. The clinical use of all debridement methods within 120 s was selected according to previously reported studies of Er:YAG lasers. Since the Er,Cr:YSGG laser is less efficient when compared to the Er:YAG laser, the Er,Cr:YSGG could give different results in a study design where the application time is not limited or longer. Another reason why Er,Cr:YSGG is less effective may be that the 2.940 μm wavelength of the Er:YAG laser matches the water absorption peak, while the Er,Cr:YSGG laser has an approximately three times lower absorption coefficient in water due to its 2.780 μm wavelength [33].

We evaluated the three-dimensional roughness data together with the SEM images by profilometry, and the Ra values were measured. There were fewer peaks and valleys before debridement procedures when a thicker debris layer masked the typical SLA surface of the implant. It was observed that Ra values measured from these areas increased after debridement procedures. On the contrary, the surface roughness decreased in implants where a relatively clean thread was selected at baseline. Since the hard attachments on the extracted implant surface due to peri-implantitis were not distributed homogeneously, the limited area examined does not represent the whole implant surface. It may be more reliable to measure with techniques that can display the entire implant surface area instead of linear values (Ra values) of a chosen spot.

## 5. Conclusions

The topographic and elemental evaluation of the surface change with SEM, EDX, and profilometry methods as a result of debridement with erbium lasers (Er:YAG and Er,Cr: YSGG) and mechanical debridement methods (titanium curette, ultrasonic device) in implants that have been removed due to peri-implantitis resulted in the following:

ErL-LP was the most efficient in debriding the implant without damaging the surface. Besides a few particles left on the implant surface, US-PEEK was effective as well. ErL-SP and ErL-VSP interventions were also efficient in terms of cleanness, but some surface damage was seen. Ti-Cur could not achieve a thorough cleaning and resulted in some surface scratching. ErCrL was ineffective in this specific application duration and energy density.

## Figures and Tables

**Figure 1 nanomaterials-11-02602-f001:**
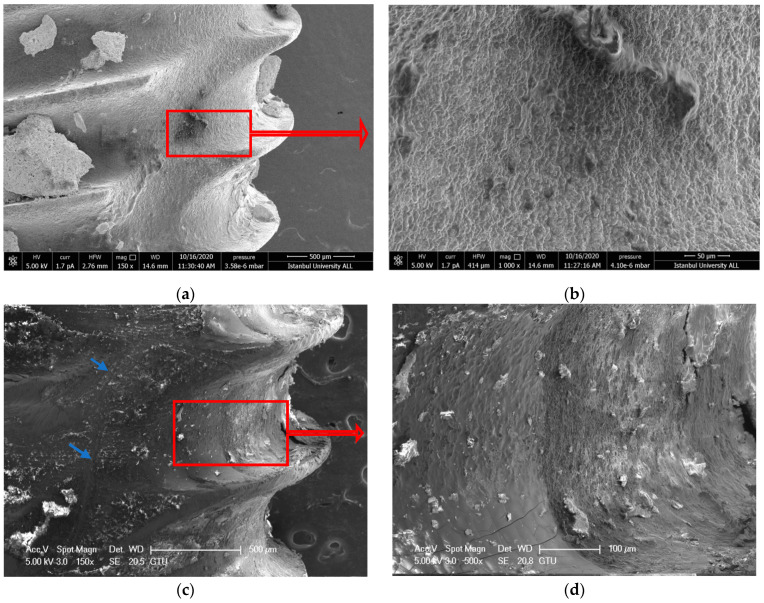
(**a**,**b**) SEM images showing the topography of untreated failed implant surfaces in Ti-Cur group (150× and 1000×) (**c**,**d**) after intervention (150× and 1000×). The blue arrows point to scratched areas. Red arrows show the further magnification area.

**Figure 2 nanomaterials-11-02602-f002:**
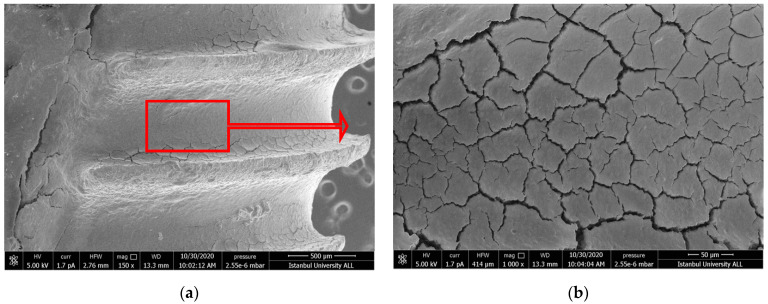
(**a**,**b**) SEM images showing the topography untreated failed implant surfaces in US-PEEK group (150× and 1000×). The image in (**b**) represents the contamination layer on the implant surface in all SEM images; (**c**,**d**) after intervention (150× and 1000×). The blue arrows point to PEEK material remnants. Red arrows show the further magnification area.

**Figure 3 nanomaterials-11-02602-f003:**
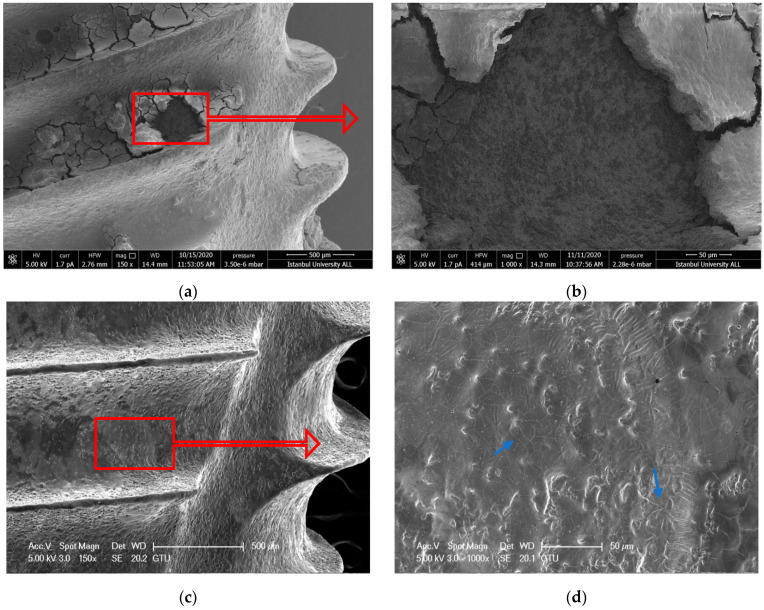
(**a**,**b**) SEM images showing the topography of untreated failed implant surfaces in ErL-VSP group (150× and 1000×) (**c**,**d**) after intervention (150× and 1000×). The blue arrows point to microcracks. Red arrows show the further magnification area.

**Figure 4 nanomaterials-11-02602-f004:**
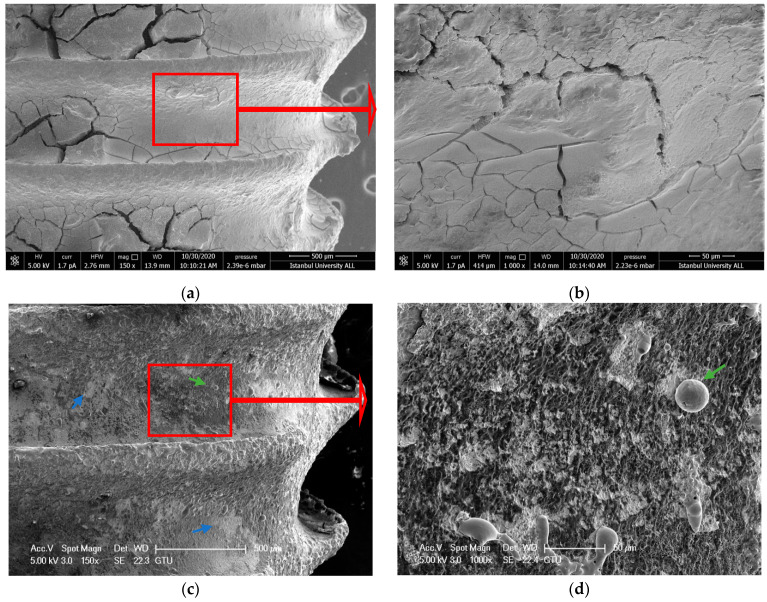
(**a**,**b**) SEM images showing the topography of untreated failed implant surfaces in ErL-SP group (150× and 1000×) (**c**,**d**) after intervention (150× and 1000×). The blue arrows indicate delamination and the green arrows indicate melting. Red arrows show the further magnification area.

**Figure 5 nanomaterials-11-02602-f005:**
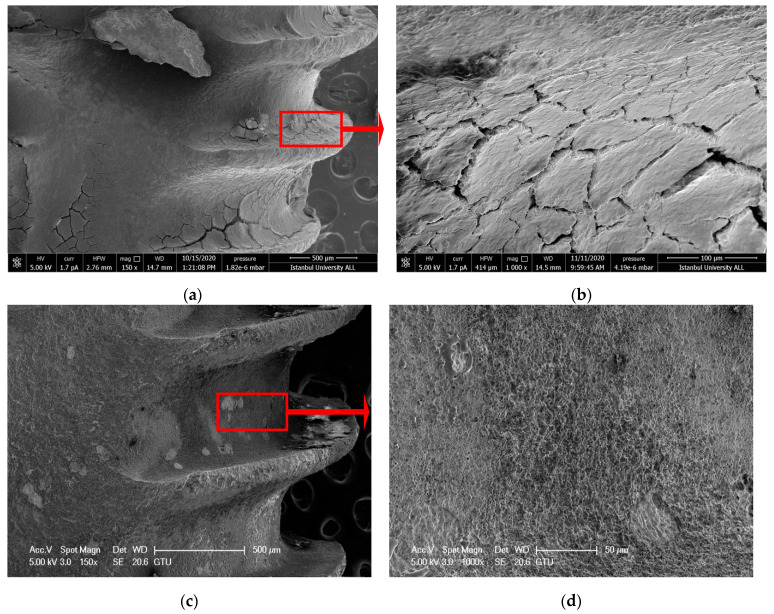
(**a**,**b**) SEM images showing the topography of untreated failed implant surfaces in ErL-LP group (150× and 1000×) (**c**,**d**) after intervention (150× and 1000×). Red arrows show the further magnification area.

**Figure 6 nanomaterials-11-02602-f006:**
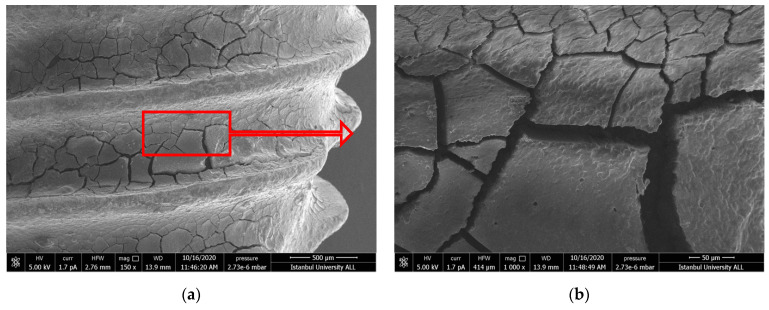
(**a**,**b**) SEM images showing the topography of untreated failed implant surfaces in ErCrL-1 group (150× and 1000×); (**c**,**d**) after intervention (150× and 1000×). Red arrows show the further magnification area.

**Figure 7 nanomaterials-11-02602-f007:**
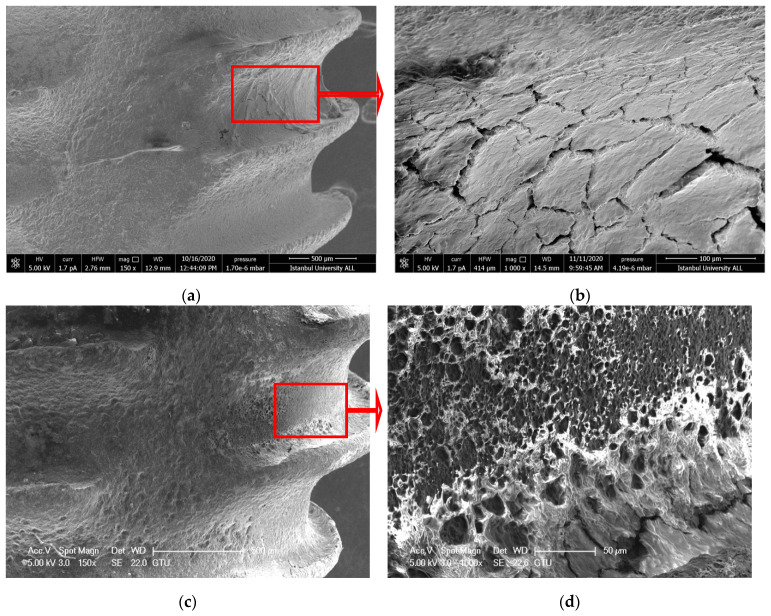
(**a**,**b**) SEM images showing the topography of untreated failed implant surfaces in ErCrL-2 group (150× and 1000× magnification); (**c**,**d**) after intervention (150× and 1000× magnification). Red arrows show the further magnification area.

**Figure 8 nanomaterials-11-02602-f008:**
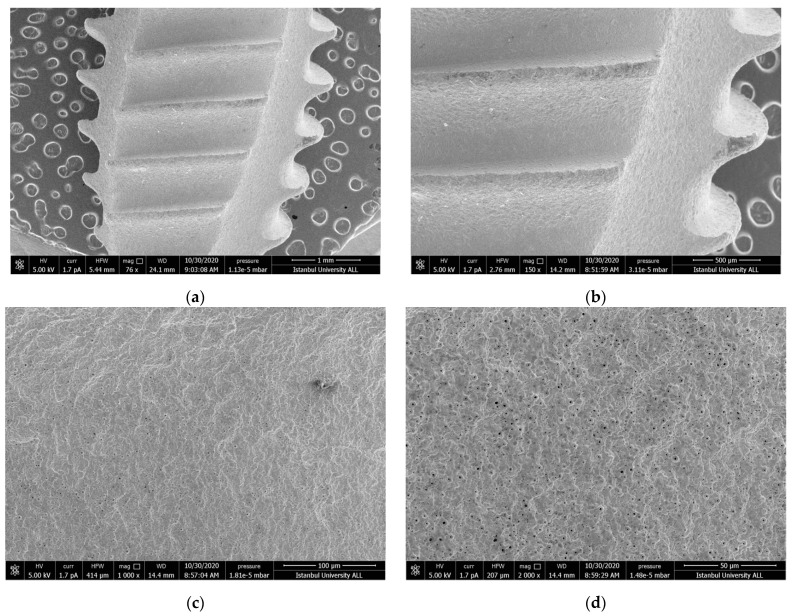
SEM images showing the topography original surface of virgin implant (**a**) 76×; (**b**) 150×; (**c**) 1000×; (**d**) 2000× magnification.

**Figure 9 nanomaterials-11-02602-f009:**
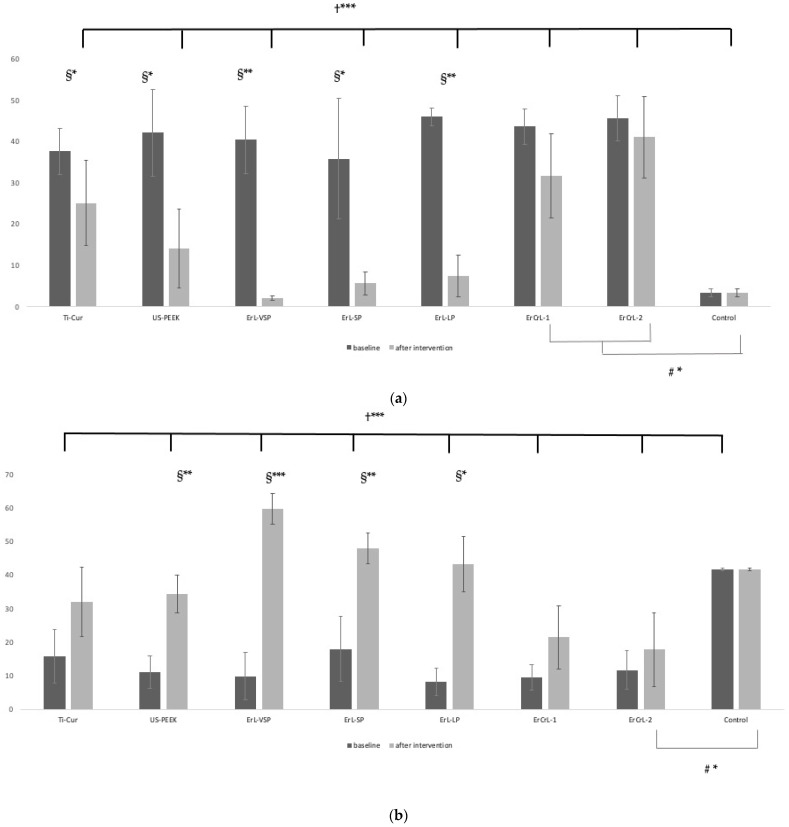
(**a**) EDX % C intragroup and intergroup comparisons; (**b**) EDX % Ti intragroup and intergroup comparisons; (**c**) EDX % O intragroup and intergroup comparisons; (**d**) EDX % N intragroup and intergroup comparisons. §: Paired Sample *t* Test †: One-Way ANOVA #: POST HOC Tukey HSD *p*-value * < 0.05 ** < 0.01 *** < 0.001.

**Figure 10 nanomaterials-11-02602-f010:**
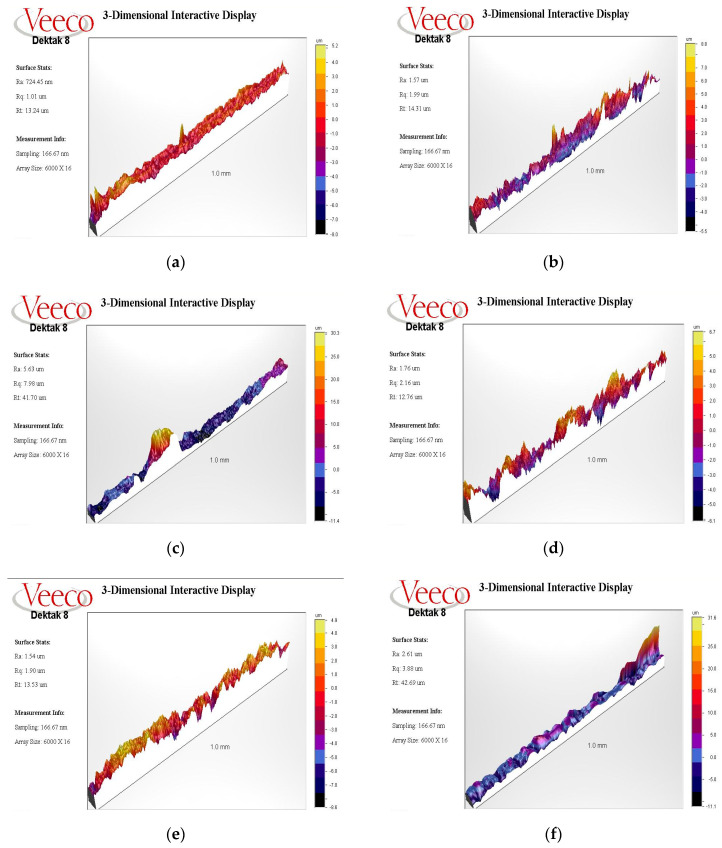
Profilometry graphic sample of the Ti-Cur group (**a**) Ra score baseline 724.45 nm; (**b**) after intervention 1.57 µm; (**c**) The US-PEEK group Ra score baseline 5.63 µm; (**d**) after intervention 1.76 µm, In the baseline measurement, a large debris (30.3 µm) in a relatively small area was observed to increase the mean roughness (Ra 5.63 µm). Although an increase in roughness was observed in the graph after intervention, the average roughness value decreased (Ra 1.76 µm); (**e**) the ErL-VSP group Ra score baseline 1.54 µm; (**f**) after intervention 2.61 µm, Although the yellow color scale in the initial measurement graph of this sample turned into a dark color scale due to the reduction of the debris layer in the thread, debris at one point peaked in the graph; (**g**) the ErL-SP group Ra scores baseline 1.82 µm; (**h**) after intervention 1.44 µm; (**i**) The ErL-LP group Ra score baseline 3.30 µm; (**j**) after intervention 1.22 µm, in all implant samples from ErL-SP and ErL-LP groups, the roughness graph of after debridement measurement shows a closer look to the image of control implants than the initial measurement; (**k**) the ErCrL-1 group Ra score baseline 1.39 µm; (**l**) after intervention 1.30 µm; (**m**) the ErCrL-2 group Ra score baseline 868.14 nm; (**n**) after intervention 1.64 µm, in ErCrL-2 group implants, a homogeneous peak-to-valley distribution cannot be observed in the roughness graph after debridement; (**o**) the control group Ra scores 731.12 nm; (**p**) 1.37 µm.

**Table 1 nanomaterials-11-02602-t001:** Treatment procedures.

Groups	Instruments	Parameters
Ti-Cur (*n* = 4)	Titanium curette ^1^	120 s
US-PEEK (*n* = 4)	Ultrasonic scaler ^2^ with PEEK ^3^ tip	120 s
ErL-VSP (*n* = 4)	ER:YAG laser ^4^ with R02-C	VSP (100 µs); 120 mJ/pulse; 10 Hz; Air 6 Water 4; 120 s; 19.04 J/cm^2^
ErL-SP (*n* = 4)	ER:YAG laser ^4^ with R02-C	SP (300 µs); 120 mJ/pulse; 10 Hz; Air 6 Water 4; 120 s; 19.04 J/cm^2^
ErL-LP (*n* = 4)	ER:YAG laser ^4^ with R02-C	LP (600 µs); 120 mJ/pulse; 10 Hz; Air 6 Water 4; 120 s; 19.04 J/cm^2^
ErCrL-1 (*n* = 4)	Er,Cr:YSGG laser ^5^ with RFPT5 14 mm fiber tip	1 W, 10 Hz (100 mJ/pulse); Air 40 Water 50; 120 s; 38.46 J/cm^2^
ErCrL-2 (*n* = 4)	Er,Cr:YSGG laser ^5^ with RFPT5 14 mm fiber tip	1.5 W, 30 Hz (50 mJ/pulse); Air 40 Water 50; 120 s; 19.23 J/cm^2^
Control (*n* = 2)	No intervention was made on the implants	

^1^ LM ErgoMix^TM^, Pargas, Finland. ^2^ Woodpecker^®^, Guilin, China. ^3^ Scorpion^TM^ İnsert CLiP Fine Ultrasonic Implant Scaler Kit^TM^, Romagnat, France. ^4^ Fotona^®^ Fidelis Plus II, Ljubljana, Slovenia. ^5^ WaterLase iPlus^®^, Foothill Ranch, CA, USA.

**Table 2 nanomaterials-11-02602-t002:** Implant debridement visual index scores.

Group	O1: DR/DnR	O2: DR/DnR	O3: DR/DnR
Ti-Cur	0/4	1/3	1/3
US-PEEK	3/1	3/1	3/1
ErL VSP	4/0	4/0	4/0
ErL SP	4/0	4/0	3/1
ErL LP	4/0	4/0	4/0
ErCrL-1	1/3	2/2	1/3
ErCrL-2	2/2	2/2	2/2

O1: observer 1, O2: observer 2, O3: observer 3, DR: debris removed, DnR: debris not removed.

## Data Availability

Data are contained within the article.

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
