# Peer review of "Evaluation of Surface Change and Roughness in Implants Lost Due to Peri-Implantitis Using Erbium Laser and Various Methods: An In Vitro Study"

_nanomaterials, 2021, doi:10.3390/nano11102602_

Round 1

Reviewer 1 Report

This is an interesting study and the authors focused on a new aspect that is poorly investigated. Therefore, this work could have clinical relevance; however, a major revision of the manuscript is required:

  1. The abstract results not discursive and poorly clear. The reader needs to see the whole manuscript to have a clear idea of the experimental design and the objectives. In particular, the aims are differently reported in the sections of the text:
  • In the abstract (line 15) the authors had as the main objective to gain similar characteristics of the post-extracted implants surfaces respect to virgin implant after a debridement, but reading the introduction (line 86), a further aim emerges that is to compare the different cleaning methods among them. On the contrary, in this part the main objective is not mentioned.
  • Moreover, within the text (lines 358-360), it is not reported the comparison among the different cleaning methods.
  1. The introduction provides sufficient background, but the topics result disconnected from each other. Although, the line 44 reported “calcified deposits” there are not previous references of them. I suggest introducing this matter. The description of the two laser tools is missing in the materials and methods section: the presence of chromium ions in the laser Er,Cr:YSGG is mentioned in the discussion, but not previously.
  2. English language of this article needs to be improved:
    • In the lines 55-60 it should be used some synonyms of “damage” word.
    • Some sentences are poorly clear (i.e. lines 62-63, 313-314, 379-381, 406-408).
    • Check the English grammar (i.e. lines 35, 229, 313, 373, 399).

  1. Although some arrows appear in the figures, there is any description of them in the captions. Furthermore, the figure 6 shows a box with arrow to indicate the higher magnification of a detail, whereas this box is lacking in the other figures. I recommend using the same method to represent all figures.
  2. The reference 5 (line 501) is not conform to the journal guidelines.

Author Response

REVIEWER 1:

This is an interesting study and the authors focused on a new aspect that is poorly investigated. Therefore, this work could have clinical relevance; however, a major revision of the manuscript is required:

  1. The abstract results not discursive and poorly clear. The reader needs to see the whole manuscript to have a clear idea of the experimental design and the objectives. In particular, the aims are differently reported in the sections of the text:
  • In the abstract (line 15) the authors had as the main objective to gain similar characteristics of the post-extracted implants surfaces respect to virgin implant after a debridement, but reading the introduction (line 86), a further aim emerges that is to compare the different cleaning methods among them. On the contrary, in this part the main objective is not mentioned.
  • Moreover, within the text (lines 358-360), it is not reported the comparison among the different cleaning methods.

We have revised the aim in the introduction part accordingly (line 86):

Our aim is to obtain similar surface properties and elemental composition to virgin implants using erbium lasers (Er:YAG and Er,Cr:YSGG) and mechanical methods (curette, ultrasonic device) in the debridement of implants lost due to peri-implantitis by evaluating the surface change of the implants treated with these methods by scanning electron microscopy (SEM), energy dispersive x-ray spectroscopy (EDX), and profilometry.

We have revised the aim in the discussion part accordingly (lines 358-360):

Our aim was to ensure that the debris-covered dirty implants salvage the surface properties and elemental composition of virgin implants without changing the surface morphology after using different debridement methods.

  1. The introduction provides sufficient background, but the topics result disconnected from each other. Although, the line 44 reported “calcified deposits” there are not previous references of them. I suggest introducing this matter. The description of the two laser tools is missing in the materials and methods section: the presence of chromium ions in the laser Er,Cr:YSGG is mentioned in the discussion, but not previously.

We have now added the following references 10,11,12 and the description of calcified deposits as follows.

The development of an adherent layer of plaque (biofilm) on the implant can result in the formation of calcified deposits similar to calculus and appears to be critical for the development of peri-implant diseases [10-12]. Therefore, the primary goal of treatment is to eliminate all microbial and calcified deposits from the surface of the implant and to supply appropriate conditions for re-osseointegration [9,13].

We have added the following information of the used lasers in the introduction part:

Different laser systems have been tested for potential use in implant surfaces in-cluding erbium-doped:yttrium, aluminum, and garnet (Er:YAG), and erbium, chro-mium-doped:yttrium, scandium, gallium, and garnet (Er,Cr:YSGG) lasers[30]. Due to the high absorption by water, the 2940 nm wavelength of Er:YAG lasers are seemed to be capable of effectively removing bacterial deposits from either smooth or rough titanium implants without damaging their surfaces [31,32]. Although both Erbium lasers have high absorption by water, the Er,Cr:YSGG laser operating at a wavelength of 2,780 nm has a lower absorption coefficient in water than the Er:YAG laser [33].

  1. English language of this article needs to be improved:
  • In the lines 55-60 it should be used some synonyms of “damage” word.
  • Some sentences are poorly clear (i.e. lines 62-63, 313-314, 379-381, 406-408).
  • Check the English grammar (i.e. lines 35, 229, 313, 373, 399).

We have revised the parts stated above in the manuscript. 

  1. Although some arrows appear in the figures, there is any description of them in the captions. Furthermore, the figure 6 shows a box with arrow to indicate the higher magnification of a detail, whereas this box is lacking in the other figures. I recommend using the same method to represent all figures.

We have now added the descriptions of arrows and the box that shows the exact place of the magnification as the reviwer suggest.

  1. The reference 5 (line 501) is not conform to the journal guidelines.

Thank you. We have corrected the reference.

Reviewer 2 Report

1. This manuscript provided information for possible clinical application and was well organized.
2. Lines 62 – 71 in the introduction seem to be more suitable in the discussion part.
3. Calcium is an important element in the oral environment and also in the biofilm, and was also reported as mentioned in the manuscript. However, “In our study, C, Ti, O, and, N elements were evaluated by EDX analysis.” The authors need to justify the reason.
4. The meaning of red arrows in the SEM images should be clarified in the captions.

Author Response

REVIWER 2:

  1. This manuscript provided information for possible clinical application and was well organized.

We appreciate the time and effort that you have dedicated to providing feedback on our manuscript.

  1. Lines 62 – 71 in the introduction seem to be more suitable in the discussion part.

We would rather keep it in the introduction part.

  1. Calcium is an important element in the oral environment and also in the biofilm, and was also reported as mentioned in the manuscript. However, “In our study, C, Ti, O, and, N elements were evaluated by EDX analysis.” The authors need to justify the reason.

While SEM helps to make a morphological evaluation of a particular surface, EDX is used to make quantitative chemical analysis by utilizing the energies of the elements. According to the EDX results in the study, elements such as C, Ti, O, N, Ca, P, K, S, Cl were found at varying rates on the implant surface. However, other elements other than these four elements (C, Ti, O, N) were not included in the statistics because they were not found in every implant examined or were found in very small amounts. According to the literature, carbon is considered as the main pollutant element and impairs biocompatibility. As we mentioned in the publication, in other studies on failed implants, according to the statistical results of EDX analysis, C decreased and Ti increased as the surface was cleaned. In addition, according to the literature, the oxygen element detected on the implant surfaces can be attributed to the titanium oxide layer, which prevents the corrosion of titanium and increases the biocompatibility , so it may have clinical significance.

  1. The meaning of red arrows in the SEM images should be clarified in the captions.

We have now added the descriptions of arrows as the reviwer suggest.

Round 2

Reviewer 1 Report

The authors adequately responded  to the reviewer's requests, thus the manuscript is accepted for the publication, although further English edits are required (i.e. lines 35, 401).